# Pregnancy Under Pressure: Oxidative Stress as a Common Thread in Maternal Disorders

**DOI:** 10.3390/life15091348

**Published:** 2025-08-26

**Authors:** Alexandru-Dan Assani, Lidia Boldeanu, Isabela Siloși, Mihail Virgil Boldeanu, Anda Lorena Dijmărescu, Mohamed-Zakaria Assani, Maria-Magdalena Manolea, Constantin-Cristian Văduva

**Affiliations:** 1Doctoral School, University of Medicine and Pharmacy of Craiova, 200349 Craiova, Romania; alexandruassani@gmail.com; 2Department of Microbiology, Faculty of Medicine, University of Medicine and Pharmacy of Craiova, 200349 Craiova, Romania; lidia.boldeanu@umfcv.ro; 3Department of Immunology, Faculty of Medicine, University of Medicine and Pharmacy of Craiova, 200349 Craiova, Romania; isabela_silosi@yahoo.com (I.S.); mihail.boldeanu@umfcv.ro (M.V.B.); 4Department of Obstetrics and Gynecology, Faculty of Medicine, University of Medicine and Pharmacy of Craiova, 200349 Craiova, Romania; magdalena.manolea@umfcv.ro (M.-M.M.); cristian.vaduva@umfcv.ro (C.-C.V.)

**Keywords:** oxidative stress, pregnancy complications, preeclampsia, gestational diabetes mellitus, fetal growth restriction, recurrent pregnancy loss, antioxidants, MDA, 8-OHdG, 8-iso-PGF2α, biomarker standardization

## Abstract

Oxidative stress, defined as the imbalance between reactive oxygen species (ROS) and antioxidant defenses, plays a pivotal role in the pathogenesis of several pregnancy complications, notably preeclampsia (PE), gestational diabetes mellitus (GDM), fetal growth restriction (FGR), and recurrent pregnancy loss (RPL). During normal pregnancy, low to moderate ROS levels support essential placental functions such as angiogenesis and trophoblast differentiation. However, excessive ROS production overwhelms antioxidant systems, leading to lipid peroxidation, protein and DNA damage, and impaired placental function. This review synthesizes current evidence linking oxidative stress to adverse pregnancy outcomes, highlighting key biomarkers such as malondialdehyde (MDA), 8-hydroxy-2′-deoxyguanosine (8-OHdG), and 8-iso-prostaglandin F2α (8-iso-PGF2α). While antioxidant therapies—particularly vitamins C and E, selenium, and folic acid—have shown promise in reducing oxidative markers, their impact on clinical outcomes remains inconsistent. The variability in results underscores the need for standardized biomarker protocols and personalized treatment strategies based on genetic predispositions and baseline oxidative status. Future research may better harness antioxidant interventions to improve maternal–fetal health by addressing these gaps.

## 1. Introduction

Oxidative stress occurs when there is an imbalance between reactive oxygen species (ROS) production and the body’s antioxidant defense, resulting in cellular damage. ROS, such as superoxide anions, hydrogen peroxide, and hydroxyl radicals, are natural by-products of cellular metabolism. Under normal conditions, antioxidants neutralize ROS to maintain cellular stability. However, when ROS production surpasses the capacity of antioxidants, oxidative stress happens, causing damage to cellular lipids, proteins, and deoxyribonucleic acid (DNA) [1,2]. Figure 1 shows the balance between ROS and antioxidant defenses under normal conditions. When this balance is disturbed, oxidative stress happens, resulting in damage to cellular lipids, proteins, and DNA. During pregnancy, increased ROS production—especially in the placenta due to higher metabolic activity and oxygen changes—may lead to problems such as preeclampsia (PE), gestational diabetes mellitus (GDM), fetal growth restriction (FGR), and recurrent pregnancy loss (RPL).

During pregnancy, the impact of ROS on cellular components is especially signifi-cant due to the numerous physiological and metabolic changes that occur. These changes involve increased oxygen consumption, heightened mitochondrial activity, and increased blood flow to the placenta, all of which contribute to elevated ROS pro-duction. The placenta, a highly vascular and metabolically active organ, becomes a main site of ROS generation because of its high energy needs and varying oxygen levels. Evidence suggests that a low level of oxidative stress is not only unavoidable but may also be necessary for normal placental development. Specifically, moderate levels of ROS are involved in essential processes like trophoblast differentiation, blood vessel remodeling, and angiogenesis– the formation of new capillaries from existing blood vessels, crucial for tissue growth, including the placenta– via redox-sensitive signaling pathways. These controlled oxidative signals support proper implantation, spiral artery transformation, and placental blood vessel growth. However, if ROS production surpasses the antioxidant defense system’s capacity, the balance is lost, causing harmful oxidative stress that can impair placental function and lead to adverse pregnancy outcomes [3,4,5,6,7]. Nevertheless, excessive ROS production disrupts this balance, leading to damage in the following ways [6,8,9,10,11,12,13]:ROS target polyunsaturated fatty acids in cellular membranes through lipid peroxidation, generating lipid radicals and end-products like malondialdehyde (MDA) that damage membrane integrity, increase cell permeability, and promote inflammation. In pregnancy, increased lipid peroxidation signifies oxidative stress, linked to conditions such as PE and FGR;Proteins can be altered by ROS, affecting their structure and function. This can lead to a loss of enzyme activity, disrupted signaling, and impaired cellular processes. For instance, oxidative damage to antioxidant enzymes like superoxide dismutase (SOD) and catalase can reduce the cell’s ability to neutralize ROS, starting a self-perpetuating cycle of oxidative damage;ROS can cause oxidative damage to DNA, leading to strand breaks, base modifications, and cross-linking, which can result in mutations and impaired cellular function. This DNA damage, caused by ROS, contributes to placental aging during pregnancy and has been linked to complications like RPL.

Research demonstrates that adverse obstetric outcomes—including PE, GDM, FGR, and RPL—are frequently correlated with elevated biomarkers of oxidative stress such as MDA and 8-hydroxy-2′-deoxyguanosine (8-OHdG), as well as reduced antioxidant capacity in maternal serum and placental tissues [14,15,16]. By elucidating the interactions between reactive oxygen species and various cellular constituents, we can deepen our understanding of oxidative stress mechanisms contributing to pregnancy pathologies, thereby informing biomarker discovery and therapeutic strategies [17,18,19].

## 2. Materials and Methods

### 2.1. Search Strategy

A comprehensive literature search was performed using PubMed, targeting studies elucidating the mechanisms of oxidative stress in pregnancy-related complications. The search strategy employed a combination of keywords including “oxidative stress,” “pregnancy complications,” “preeclampsia,” “gestational diabetes mellitus,” “fetal growth restriction,” and “antioxidant therapy”. The search was restricted to peer-reviewed articles published from 200 2025. Figure 2 depicts the methodology underlying the search strategy [20].

### 2.2. Inclusion Criteria and Exclusion Criteria

Research studies were chosen according to the following criteria for inclusion:Research Type: Original studies, clinical trials, and systematic reviews in peer-reviewed journals;Population: Research on human participants, particularly pregnant women with complications related to oxidative stress, including preeclampsia, gestational diabetes, fetal growth restriction, or recurrent pregnancy loss;Content: Research on oxidative stress biomarkers, mechanisms, and antioxidant therapies, as well as clinical outcomes related to oxidative stress during pregnancy;Publication Date: Studies published within the past 25 years (2000–2025).Research studies were eliminated based on specific criteria:Irrelevant Outcomes: Articles that focus on oxidative stress in non-pregnancy settings or in populations other than pregnant women;Incomplete Data: Studies that lack sufficient information on oxidative stress markers, clinical outcomes, or intervention details;Non-Human Studies: Research conducted exclusively on animals or in vitro, without direct relevance to human clinical applications;Non-Original Research: We excluded editorials, opinion pieces, and case reports.

## 3. Understanding Oxidative Stress Mechanisms During Pregnancy

Biochemical pathways mediating oxidative stress during pregnancy primarily involve augmented ROS generation, lipid peroxidation, and exhaustion of antioxidant defenses. Mitochondria serve as a principal ROS source, with increased bioenergetic demands during gestation—especially within placental tissues—contributing to excessive ROS production. If antioxidant mechanisms are inadequate, this imbalance can precipitate cellular and tissue damage [21,22,23].

### 3.1. The Connection Between Lipid Peroxidation and Pregnancy-Related Hypertension

Lipid peroxidation, a critical oxidative process, greatly contributes to the development of pregnancy-related hypertensive disorders, including PE. During lipid peroxidation, ROS attack polyunsaturated fatty acids in the phospholipid bilayer of cell membranes, initiating a chain reaction that produces lipid radicals and reactive aldehydic by-products such as MDA and 4-hydroxynonenal (4-HNE). These electrophilic compounds alter membrane structure, increase cellular permeability, and act as signaling mediators that boost oxidative stress, inflammation, and endothelial dysfunction [3,24,25].

In pregnancy-associated hypertension, increased lipid peroxidation in placental and vascular endothelial cells impairs cell function, thereby contributing to the development of multiple adverse outcomes. Endothelial dysfunction, characterized by increased lipid peroxidation, causes injury to endothelial cells, which decreases nitric oxide (NO) bioavailability. NO, a crucial endothelium-derived relaxing factor, facilitates vasodilation; when its production or activity is reduced due to oxidative stress, it leads to vasoconstriction, increased systemic vascular resistance, and the hypertensive condition observed in preeclampsia. Lipid peroxidation derivatives such as 4-HNE act as potent pro-inflammatory mediators, inducing cytokine release that worsens vascular inflammation. This process worsens endothelial damage, leading to impaired placental blood flow and increased oxidative stress, thus creating a self-sustaining pathological cycle. Placental hypoxia occurs when decreased uteroplacental blood flow, caused by vasoconstriction, creates a hypoxic environment. This condition triggers ROS production through hypoxia-reoxygenation mechanisms, which further heighten oxidative stress and lipid peroxidation. The oxidative damage to placental tissues from this process contributes to obstetric conditions such as PE and FGR [3,26,27,28,29,30,31].

Clinical evidence highlights the significant increase in lipid peroxidation biomarkers, especially MDA levels, in women with preeclampsia compared to normotensive pregnant controls. These findings support the idea that lipid peroxidation is both a reliable marker of oxidative stress and a mechanistic factor in the development of hypertensive disorders during pregnancy, emphasizing its potential role in the cause and development of targeted treatments [32,33,34].

### 3.2. Advances in Clinical Insights on Angiogenic Imbalance and Oxidative Biomarkers in Preeclampsia

Contemporary prospective data highlight the complex relationship between oxidative stress biomarkers and angiogenic imbalance in preeclampsia. The study tracked levels of soluble fms-like tyrosine kinase-1 (sFlt-1), placental growth factor (PlGF), soluble endoglin (sEng), and 8-epi-prostaglandin F2-alpha (8-epi-PGF2α) at multiple points during pregnancy in women with suspected or confirmed PE. Results showed that elevated 8-epi-PGF2α, a marker of lipid peroxidation, was strongly linked to anti-angiogenic factors (sFlt-1, sEng) and the sFlt-1/PlGF ratio—a well-known diagnostic marker for PE. Notably, 8-epi-PGF2α levels increased proportionally with gestational age, mirroring rises in sFlt-1 and decreases in PlGF. This pattern indicates a growing angiogenic imbalance and a rise in oxidative stress, emphasizing the role of redox imbalance in the development of PE. Additionally, 33.3% of women with a baseline sFlt-1/PlGF ratio (≤38) later experienced an increase above the diagnostic cutoff (>38), highlighting the importance of dynamic biomarker monitoring. The study recommends including 8-epi-PGF2α alongside angiogenic markers for early PE detection and risk assessment [35].

### 3.3. The Relationship Between 8-iso-Prostaglandin F2α and Preeclampsia

Serum 8-iso-PGF2α, a reliable marker of lipid peroxidation, was significantly higher in women with PE compared to normotensive pregnant women. Postpartum levels decreased but stayed elevated compared to healthy pregnancies, indicating ongoing oxidative stress. 8-iso-PGF2α levels showed strong positive correlations with systolic (SBP) and diastolic blood pressure (DBP), tumor necrosis factor alpha (TNF-α) (a pro-inflammatory cytokine), pentraxin 3 (PTX3) (a marker of vascular inflammation), and a moderate correlation with interleukin 10 (IL-10), an anti-inflammatory cytokine. Receiver operating curve (ROC) analysis showed that 8-iso-PGF2α had 93.3% sensitivity and 96.7% specificity (AUC = 0.949), indicating high diagnostic accuracy for PE. PTX3 and IL-6 also had diagnostic value, though with lower specificity. These findings support using 8-iso-PGF2α and PTX3 as early predictive biomarkers for PE, which may help enable earlier diagnosis and preventative measures, improving maternal-fetal outcomes [36].

### 3.4. Lifestyle Factors Influencing Oxidative Stress During Pregnancy

Nutrition significantly influences oxidative stress levels during pregnancy. Diets high in polyphenols, omega-3 fatty acids, vitamins C and E, and other antioxidants can decrease oxidative damage and boost antioxidant defenses. Such diets are linked to lower the levels of oxidative biomarkers such as MDA and 8-OHdG in both maternal serum and placental tissues [37]. Deficiencies in micronutrients like selenium, folate, and vitamin C are associated with increased oxidative stress and a higher risk of conditions like PE and FGR [38].

Cigarette smoke contains a high level of free radicals and pro-oxidant compounds. In pregnant smokers, this exposure causes increased lipid peroxidation and reduces antioxidants such as glutathione and vitamin C [13]. Smoking is also independently linked to increased oxidative DNA damage and has been involved in the development of RPL, placental insufficiency, and fetal hypoxia—conditions where oxidative stress is vital [12].

Alcohol intake during pregnancy causes mitochondrial dysfunction and increased ROS production, mainly by reducing glutathione synthesis and disrupting folate metabolism. This can result in oxidative damage to the placenta, fetal alcohol spectrum disorders, growth restriction, and miscarriage. Alcohol also hampers absorbing essential antioxidants such as folate, zinc, and selenium, further weakening cellular defenses [13,39].

Regular, moderate-intensity physical activity promotes the upregulation of endogenous antioxidant enzymes like SOD and catalase, and it has been shown to reduce oxidative damage in maternal tissues [40]. In conditions such as GDM, structured exercise programs improve placental perfusion and oxidative profiles, supporting their inclusion in prenatal care protocols [41]. A review from 2025 emphasized how environmental factors like poor diet and pollution increase oxidative stress, which contributes to GDM pathogenesis [42].

### 3.5. Additional Pathways Associated with Oxidative Stress

Besides lipid peroxidation, oxidative stress also damages cellular proteins and DNA, which further contributes to pregnancy complications. ROS can alter vital proteins, affecting enzyme functions and cellular signaling pathways. For example, oxidative changes to key antioxidant enzymes like SOD reduce their catalytic activity, weakening the cell’s antioxidant defenses. This creates a positive feedback loop that worsens oxidative stress, especially in the placenta, where antioxidant defenses are already vulnerable. Oxidative stress causes various types of DNA damage, including single- and double-strand breaks and base modifications, which disrupt genomic integrity and may lead to mutations. In obstetric cases, particularly in placental disease, these DNA damages have been linked to accelerated placental aging and RPL [3,13,43,44,45].

Understanding these mechanisms—especially how lipid peroxidation contributes to hypertensive disorders in pregnancy—highlights the importance of targeted antioxidant therapies that regulate oxidative pathways. Such interventions could help improve endothelial dysfunction in high-risk pregnancies and reduce the occurrence of pregnancy-related complications caused by oxidative stress.

## 4. Pregnancy Complications Associated with Oxidative Stress

### 4.1. Preeclampsia

PE, a significant hypertensive disorder that affects approximately 8% of all pregnancies, has been extensively linked to the phenomenon of oxidative stress. Pathophysiological studies reveal that individuals diagnosed with PE exhibit markedly elevated levels of oxidative biomarkers, such as MDA and isoprostanes. Concurrently, these individuals show a reduction in the activity of endogenous antioxidant enzymes, which play a crucial role in mitigating oxidative damage. This dual presentation of heightened oxidative markers alongside decreased antioxidant defense mechanisms underscores the pivotal role that oxidative stress may play in the etiology and progression of PE [46,47,48,49,50]. The pathophysiological mechanisms of PE are predominantly associated with aberrant placental implantation and chorionic villous development, leading to compromised uteroplacental perfusion. This hypoperfusion induces episodes of hypoxia-reoxygenation injury, which in turn elevates ROS generation and promotes lipid peroxidation. These oxidative stress mechanisms contribute to endothelial dysfunction and systemic inflammatory responses characteristic of the disease [51,52,53]. Evidence indicates that preeclamptic women have significantly lower levels of vitamins C and E, highlighting a potential antioxidant deficiency in this population [54,55].

Although antioxidant interventions have been investigated for attenuating oxidative stress and enhancing clinical outcomes, the results from randomized trials remain heterogeneous. While supplementation with antioxidants such as vitamins C and E has demonstrated reductions in oxidative biomarkers, these interventions have not reliably prevented the development of PE. This underscores the necessity for more precise therapeutic strategies or optimized patient stratification [56,57,58].

### 4.2. Gestational Diabetes Mellitus

GDM is a disorder characterized by glucose intolerance that manifests during gestation, impacting both maternal and fetal outcomes. Oxidative stress plays a critical role in the pathophysiology of GDM by promoting inflammatory signaling cascades and inducingβ-cell dysfunction [59,60,61]. Recent studies indicate that women with GDM exhibit elevated MDA levels and diminished activities of antioxidant enzymes, including SOD and glutathione peroxidase (GPX), in comparison to normoglycemic pregnant controls [59,62,63]. For example, Ruiz-Martínez et al. found that elevated MDA levels and body mass index (BMI) at baseline were significantly associated with postpartum hyperglycemia in women with GDM [64]. Dietary sugar intake also played a role. This imbalance extends its detrimental effects beyond maternal health, significantly impairing fetal development by inducing oxidative stress that disrupts placental function. Such disruptions can contribute to macrosomia and elevate the risk of subsequent metabolic disorders in the offspring [65,66].

Interventions aimed at modulating oxidative stress in GDM exhibit heterogeneous results. While certain studies demonstrate changes in oxidative biomarkers following antioxidant supplementation, the translation of these biochemical effects into consistent improvements in clinical outcomes for both mother and fetus remains unsubstantiated, underscoring the necessity for more refined, targeted therapeutic strategies [59,67,68].

### 4.3. Fetal Growth Restriction

FGR is characterized by a failure of the fetus to attain its genetically predetermined growth trajectory, primarily attributable to placental insufficiency. Oxidative stress has been identified as a pivotal pathogenic factor in FGR, mediating placental cellular damage and impairing transplacental transfer of nutrients and oxygen to the fetus [69,70,71]. Current research indicates that oxidative stress markers such as MDA and 8-OHdG are significantly upregulated in placental tissue and umbilical cord blood samples from FGR cases, suggesting a potential role of ROS in disrupting signaling pathways essential for normative fetal development [72].

In cases of FGR, the placental tissue frequently demonstrates histopathological features indicative of hypoxic injury and augmented lipid peroxidation. The activity of endogenous antioxidant enzymes is typically diminished, potentially heightening the susceptibility of both the placenta and fetus to oxidative stress-mediated damage. Current evidence advocates for the exploration of targeted antioxidant interventions to enhance placental function and fetal prognosis; however, further rigorous research is essential to substantiate their clinical efficacy [70,73,74,75].

### 4.4. Recurrent Pregnancy Loss

RPL, marked by two or more consecutive miscarriages, has been linked to increased oxidative stress biomarkers. Patients with RPL often show higher levels of oxidative markers such as MDA and 8-OHdG, indicating lipid peroxidation and DNA oxidation, respectively [13,76]. Reduced antioxidant activity is also observed in RPL cases, suggesting a decreased ability to neutralize ROS within reproductive tissues [13].

The potential of antioxidant therapy to mitigate oxidative stress in women with RPL has garnered considerable interest. Existing studies indicate improvements in oxidative biomarkers and a concomitant reduction in miscarriage rates following antioxidant supplementation. Nonetheless, further rigorous clinical trials are requisite to conclusively establish efficacy and optimize dosing protocols [77,78].

Lin et al. found a strong association between oxidative/nitrosative stress markers and RPL, suggesting these pathways contribute to early pregnancy failure [79]. Câmara et al. examined oxidative stress in sperm from partners of RPL patients. Results suggested that male factors may play a role in RPL through oxidative damage to sperm DNA [80]. A recent review identified ferroptosis (iron-dependent oxidative cell death) as a novel mechanism in RPL, driven by disrupted antioxidant systems and increased lipid peroxidation in the placenta [81].

The complex interplay of oxidative stress in modulating maternal and fetal outcomes is depicted in Figure 3, delineating the specific oxidative pathways involved in PE, GDM, FGR, and RPL.

## 5. Interventions for Antioxidant Support During Pregnancy

Given the critical involvement of oxidative stress in the pathogenesis of diverse pregnancy complications, numerous studies have explored the efficacy of antioxidant therapies as potential therapeutic interventions. These strategies are designed to scavenge ROS, inhibit oxidative cellular damage, and enhance pregnancy outcomes.

### 5.1. Commonly Studied Antioxidants

Vitamins C and E are extensively studied antioxidants in obstetric research. As a hydrophilic antioxidant, vitamin C mitigates ROS in the extracellular matrix, whereas vitamin E, a lipophilic antioxidant, safeguards cellular membranes from lipid peroxidation. Clinical trials have demonstrated variable effects of these vitamins on oxidative stress biomarkers; however, evidence regarding their efficacy in preventing hypertensive disorders such as PE remains inconclusive [82,83,84].

Selenium is a critical cofactor for GPX, a pivotal antioxidant enzyme. Evidence suggests that selenium supplementation can augment antioxidant defenses, particularly in populations with selenium deficiency. Nevertheless, current data regarding selenium’s role in mitigating oxidative stress–associated pregnancy complications remain limited and inconclusive [85,86,87].

Folic acid modulates oxidative stress indirectly through the reduction of homocysteine (Hcy) levels, which are associated with elevated oxidative damage. While its prophylactic role in neural tube defect (NTD) prevention is well established, the specific impact of folic acid on oxidative stress-mediated obstetric complications warrants further investigation [88,89].

### 5.2. Evidence-Based Clinical Studies on Antioxidant Therapy

Several studies investigating antioxidant therapy in PE have yielded mixed results. While several trials demonstrate a reduction in oxidative stress markers following supplementation with vitamins C and E, these interventions have not consistently demonstrated a prophylactic effect against the development of PE. These findings suggest that antioxidants may primarily mitigate disease severity rather than prevent onset, especially in high-risk cohorts [90,91,92].

Antioxidant interventions in GDM are designed to mitigate oxidative stress within the pancreatic and placental tissues. Although certain studies demonstrate modifications in oxidative biomarkers following supplementation, definitive clinical outcomes for maternal and fetal health remain to be established, underscoring the need for further investigational efforts [59,67,68]. A 2025 meta-analysis showed that selenium, alpha-lipoic acid, zinc, and epigallocatechin 3-gallate (EGCG) supplementation significantly improved insulin resistance markers (homeostatic model assessment for insulin resistance (HOMA-IR); HOMA for β cell function (HOMA-B); quantitative insulin-sensitivity check index (QUICKI)) in GDM patients [68].

Considering the established association between oxidative stress and RPL pathophysiology, antioxidant therapy has been explored as a potential prophylactic intervention. Preliminary studies suggest a reduction in miscarriage rates with antioxidant supplementation; however, high-quality, large-scale randomized controlled trials are necessary to definitively validate its efficacy as a standard treatment modality [93,94].

The heterogeneity in outcomes associated with antioxidant therapy underscores the potential necessity for personalized treatment regimens. Current research is focused on developing targeted antioxidant interventions tailored to specific patient cohorts and gestational stages. For instance, early administration during placental development may yield greater efficacy, particularly in high-risk populations. Moreover, the identification of robust biomarkers for oxidative stress could facilitate precision in therapeutic monitoring and adjustments, thereby optimizing clinical outcomes [6,23,95].

Emerging evidence underscores the significance of modifiable lifestyle factors—namely, diet and physical activity—in modulating oxidative stress. Nutritional interventions enriched with antioxidants such as polyphenols and omega-3 polyunsaturated fatty acids (PUFAs) have demonstrated efficacy in attenuating oxidative damage, thereby potentially reducing the incidence of obstetric complications like PE and FGR [37,40,41].

Personalized antioxidant therapies are increasingly being explored, with a focus on genetic polymorphisms affecting individual responses. Variants in genes such as SOD2 and GPX1, which encode critical antioxidant enzymes, may modulate the efficacy of oxidative stress mitigation, thereby guiding the development of more precise, genotype-informed therapeutic strategies [96,97,98].

## 6. Recognizing Research Gaps and the Importance of Standardizing Biomarkers

### 6.1. Standardizing Oxidative Stress Biomarkers for Improved Clinical Diagnosis

A significant obstacle to the advancement of antioxidant therapeutic strategies in pregnancy is the heterogeneity in the measurement of oxidative stress biomarkers across different studies. Although established biomarkers such as MDA, 8-OHdG, and advanced oxidation protein products (AOPPs) are routinely utilized to quantify oxidative damage, the lack of a standardized assay protocol impedes the comparability of data and the establishment of definitive thresholds for clinical significance [6,99,100].

MDA levels are frequently elevated in the context of PE and other gestational complications. However, the absence of standardized measurement protocols for MDA results in disparate cutoff thresholds across studies, complicating clinical interpretation [6,101,102]. Implementation of standardized protocols for oxidative stress biomarkers would enhance early risk stratification in high-risk pregnancies, augment the precision of risk assessments, and support the deployment of targeted interventions.

### 6.2. The Significance of Tailored Antioxidant Treatment

Another research gap pertains to delineating the specific patient cohorts that derive the maximal benefit from antioxidant therapy. Current evidence indicates that individual genetic predispositions, lifestyle factors, and baseline oxidative stress markers modulate the efficacy of antioxidants. A precision medicine approach, utilizing biomarker profiling, could optimize antioxidant interventions—enhancing outcomes in high-risk pregnancies while minimizing the risks associated with unwarranted supplementation [6,38,103].

For instance, individuals harboring polymorphisms in genes encoding antioxidant enzymes, such as SOD2 or GPX1, may necessitate tailored antioxidant interventions, including elevated dosages or specific formulations. Implementing personalized antioxidant regimens informed by standardized biomarker assessments could enhance reproductive outcomes more effectively than generalized protocols [104,105].

### 6.3. Fine-Tuning the Dose and Timing of Antioxidant Treatment

Emerging evidence indicates that the timing and dosing of antioxidants such as vitamins C and E, Se, and folic acid are pivotal in modulating their efficacy, although heterogeneity in findings persists. Variability in outcomes across studies may stem from differences in administration protocols, including dose and timing. Notably, supra-physiological doses of antioxidants have been associated with potential deleterious effects, whereas subtherapeutic doses may fail to elicit significant effects on oxidative stress pathways. Future investigations should aim to delineate optimal dosing regimens tailored to each gestational stage, with an emphasis on early pregnancy, when oxidative stress pathways integral to placental development are first engaged [106,107].

### 6.4. Future Research Recommendations

To fully leverage antioxidant therapy in pregnancy care, future research should prioritize the following areas [6,100,108,109]:Standardization of biomarker quantification is critical for ensuring reproducibility and clinical utility. Implementing uniform protocols for assessing oxidative stress biomarkers such as MDA, 8-OHdG, and AOPPs will facilitate more accurate inter-study comparisons, establish definitive oxidative stress thresholds, and improve patient stratification for targeted antioxidant therapies;Small sample sizes and heterogeneity in study methodologies predominantly constrain existing literature on antioxidant therapy in pregnancy. To establish robust evidence for efficacy, large-scale randomized controlled trials (RCTs) adopting standardized protocols and evaluating clinical endpoints—such as reductions in PE, GDM, FGR, and RPL—are essential;Tailoring antioxidant interventions based on biomarker profiles and genetic predispositions has the potential to enhance therapeutic efficacy. Future research should focus on identifying prognostic biomarkers and genetic markers capable of predicting antioxidant requirements, thereby facilitating a precision medicine approach to optimize pregnancy outcomes.

## 7. Conclusions

Oxidative stress significantly contributes to obstetric complications like PE, GDM, FGR, and RPL, involving excess ROS and weakened antioxidants that disrupt cellular functions in placental and maternal tissues. Elevated biomarkers (MDA, 8-OHdG) and reduced enzymes (SOD, GPX) are common. While antioxidant therapy makes biological sense, its clinical success varies due to complex redox signaling and patient differences.

Current evidence indicates that antioxidant supplementation—particularly with vitamins C and E, selenium, and folic acid—can enhance biochemical markers of oxidative stress. Nonetheless, these biochemical improvements do not invariably correlate with reductions in pregnancy complications, underscoring the necessity for more sophisticated and individualized therapeutic protocols. Variables such as timing, dosage, genetic predisposition (e.g., SOD2 and GPX1 polymorphisms), and baseline oxidative stress levels are critical determinants of intervention efficacy. Tailored approaches incorporating genetic screening and biomarker profiling are promising avenues for optimizing antioxidant therapy in reproductive medicine. Standardized measurement of oxidative stress biomarkers is crucial. The lack of unified protocols for markers like MDA, 8-OHdG, and AOPPs hampers comparability and hinders the development of clinical guidelines. Clear threshold levels in pregnant populations could enable earlier risk assessment and targeted treatments.

Lifestyle modifications, encompassing antioxidant-rich diets and regular physical activity, serve as adjunct non-pharmacologic interventions with the potential to mitigate oxidative stress and enhance maternal-fetal outcomes. Integration of these strategies within comprehensive care protocols may yield synergistic benefits when combined with pharmacological therapies.

Future research should focus on large-scale RCTs with standardized biomarker assessments of biochemical and clinical outcomes. Identifying optimal intervention windows, especially early pregnancy when placental development is active, may improve treatment effectiveness. Developing validated, feasible antioxidant protocols tailored to individual risks is key in managing pregnancy complications related to oxidative stress. Numerous studies lack baseline oxidative stress stratification, which causes outcome variability and hampers assessment of intervention effectiveness. Future research should include baseline oxidative profiling to account for individual differences and improve effect measurement.

In summary, although antioxidant therapy during pregnancy represents a promising therapeutic strategy, its translation into clinical practice requires further optimization. A paradigm shift towards personalized, evidence-based interventions supported by comprehensive biomarker profiling may be essential to fully realize its potential in improving maternal and fetal health outcomes. Figure 4 depicts a schematic mind map of oxidative stress mechanisms in pregnancy.

## Figures and Tables

**Figure 1 life-15-01348-f001:**
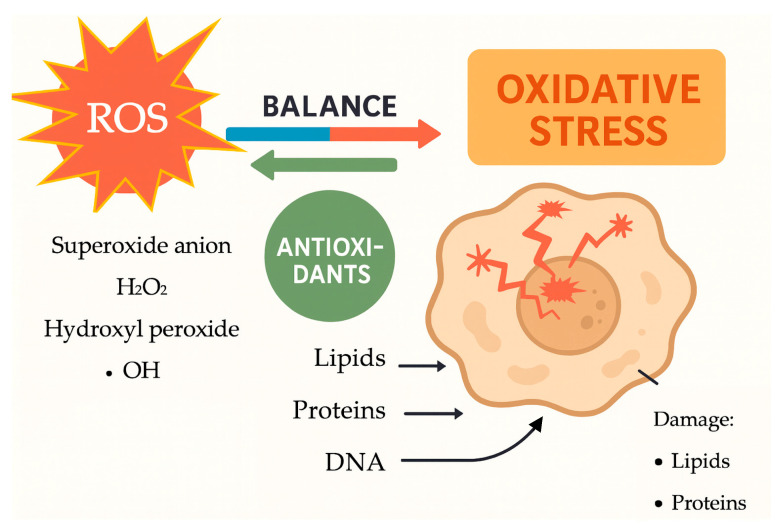
The mechanisms of oxidative stress and their effects on cellular components (Figure created in BioRender).

**Figure 2 life-15-01348-f002:**
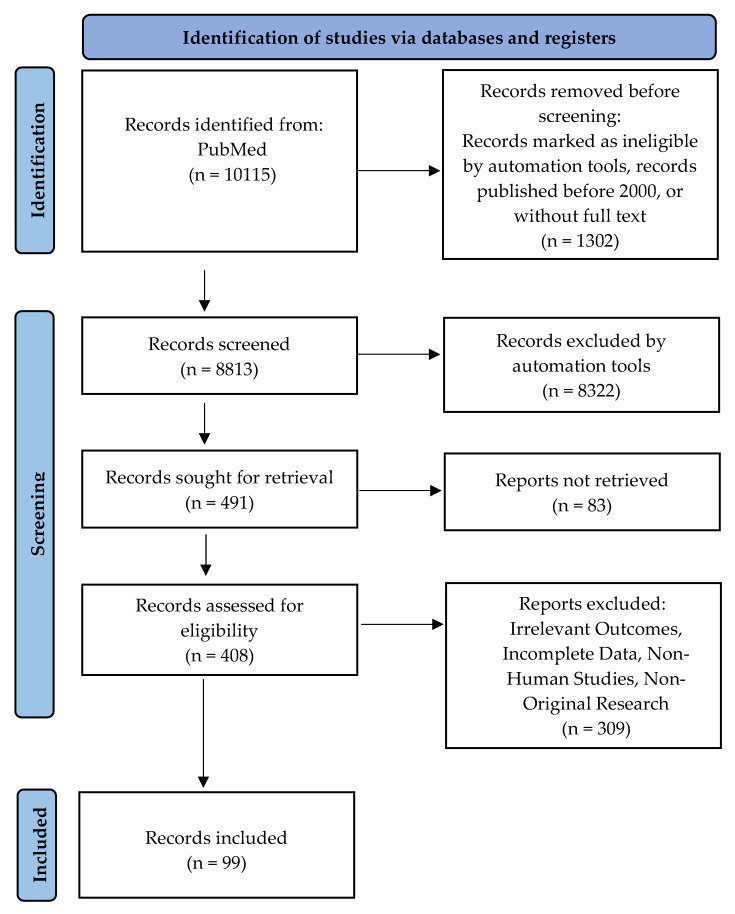
Search strategy, the PRISMA flow diagram [20].

**Figure 3 life-15-01348-f003:**
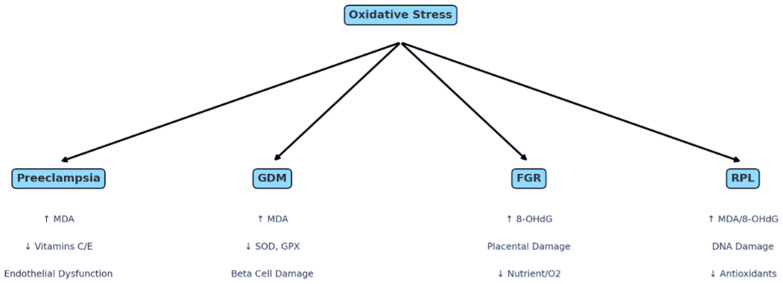
Pregnancy Complications Linked to Oxidative Stress [25,59,72,79] (Figure created in BioRender). ↑ = increase; ↓ = decrease The diagram illustrates the role of oxidative stress as a central pathogenic mechanism contributing to four major pregnancy complications: preeclampsia (PE), gestational diabetes mellitus (GDM), fetal growth restriction (FGR), and recurrent pregnancy loss (RPL). Elevated levels of reactive oxygen species (ROS) and a decline in antioxidant defenses lead to cellular damage—manifesting in distinct patterns across complications. In PE, oxidative stress is associated with increased malondialdehyde (MDA), reduced vitamins C and E, and endothelial dysfunction. GDM involves elevated MDA levels, decreased antioxidant enzymes superoxide dismutase (SOD) and glutathione peroxidase (GPX), and beta cell damage. FGR is characterized by increased 8-hydroxy-2′-deoxyguanosine (8-OHdG), placental damage, and impaired oxygen/nutrient delivery. RPL is linked to heightened oxidative DNA damage and reduced antioxidant capacity.

**Figure 4 life-15-01348-f004:**
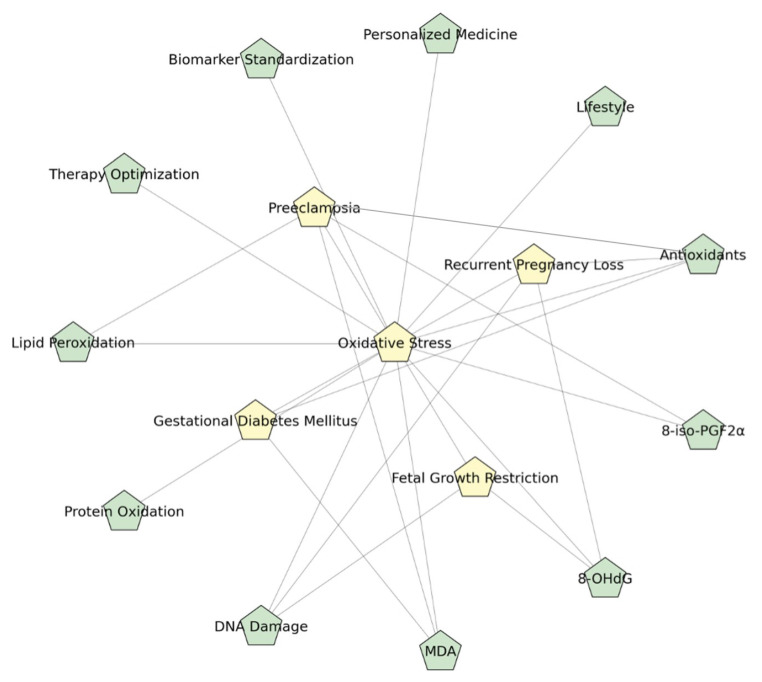
Mind Map of Oxidative Stress in Pregnancy (Figure created in BioRender). The mind map outlines the core concepts and relationships discussed in the review. At the center is oxidative stress, driven primarily by increased reactive oxygen species (ROS) and reduced antioxidant defenses. These imbalances, often amplified by mitochondrial overactivity in pregnancy, lead to cellular damage—including lipid peroxidation (e.g., malondialdehyde (MDA); 8-iso-Prostaglandin F2α (8-iso-PGF2α)), protein oxidation, and DNA damage (e.g., 8-hydroxy-2′-deoxyguanosine (8-OHdG)). Such damage is implicated in multiple pregnancy complications, including preeclampsia (PE), gestational diabetes mellitus (GDM), fetal growth restriction (FGR), and recurrent pregnancy loss (RPL). To monitor and understand oxidative stress, biomarkers such as MDA, 8-OHdG, and 8-iso-PGF2α are critical. These indicators are central to both diagnosis and therapy assessment. Current therapeutic strategies involve antioxidant supplementation (e.g., vitamins C/E, selenium, folic acid) and lifestyle interventions such as improved diet and physical activity. Future directions emphasize the need for personalized treatment, standardized biomarker protocols, and large-scale randomized controlled trials (RCTs) to refine and validate these interventions.

## Data Availability

No new data were created or analyzed in this study.

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
