# Peer review of "Pregnancy Under Pressure: Oxidative Stress as a Common Thread in Maternal Disorders"

_life, 2025, doi:10.3390/life15091348_

Round 1

Reviewer 1 Report

Comments and Suggestions for Authors

This review highlights recent advancements in oxidative stress linked to several pregnancy complications. These findings contribute to the current understanding of oxidative stress interactions and may inform future intervention approaches for this vulnerable population of pregnant women. However, certain revisions to the chapter structure are recommended to enhance the clarity and coherence of the research findings.

Even though the authors presented that imbalance of oxidative stress in more complications related to pregnancy, this review lacks recent articles regarding other complications mentioned, such as gestational diabetes mellitus and recurrent pregnancy loss.

  1. The ‘Introduction’ section is well-organized and highlights the aim of the study.

  1. Line 96-97: The search was restricted to peer-reviewed articles published up to 2025.

Please, describe the entire process of selecting the articles and the period, not only in Figure 2.

  1. For the selection process, it is recommended to use the Prisma Flow Diagram for a better understanding of how this topic has been treated.

  1. The section entitled “Mechanisms of Oxidative Stress in Pregnancy” should be revised to employ complete sentences and structured paragraphs rather than bullet points or fragmented lists.

  1. Please provide the source(s) or references used for Figure 3.

  1. Section 5 ‘Antioxidant Interventions in Pregnancy’ should be reorganized to incorporate more robust evidence from recent studies to underscore the clinical relevance of antioxidants used during pregnancy.

Furthermore, restructure into paragraphs rather than bullet points.

Author Response

Your observations have brought important clarity to several key aspects of our study, and your critical perspective has helped us refine both the structure and interpretation of our findings.

We have made every effort to address all points raised, and we have incorporated the suggested revisions wherever applicable. In cases where data or methodological limitations restricted our ability to fully implement a recommendation, we have added appropriate justifications and highlighted these as areas for future exploration. We are thrilled to any other suggestions in order to reach the best potential of our manuscript.

Thank you again for your valuable feedback and for contributing to the improvement of our manuscript.

Comment:

  1. The ‘Introduction’ section is well-organized and highlights the aim of the study.

Response: Appreciate it!

Comment:

  1. Line 96-97: The search was restricted to peer-reviewed articles published up to 2025.

Response: Updated 

Please, describe the entire process of selecting the articles and the period, not only in Figure 2.

Response: Described

Comment:

  1. For the selection process, it is recommended to use the Prisma Flow Diagram for a better understanding of how this topic has been treated.

Response: Included, if needed to be further modified, please offer your valuable opinion

Comment:

  1. The section entitled “Mechanisms of Oxidative Stress in Pregnancy” should be revised to employ complete sentences and structured paragraphs rather than bullet points or fragmented lists.

Response: We turned the bullets to paragraphs. We tried to change most of bullets to paragraphs throughout the whole review.

Comment:

  1. Please provide the source(s) or references used for Figure 3.

Response: Provided

Comment:

  1. Section 5 ‘Antioxidant Interventions in Pregnancy’ should be reorganized to incorporate more robust evidence from recent studies to underscore the clinical relevance of antioxidants used during pregnancy.

Response: We included more studies. If necessary we are opened to develop further studies for this section

Reviewer 2 Report

Comments and Suggestions for Authors

Based on the content of the article the paper identifies and tackles several critical gaps in the obstetric and maternal-fetal medicine literature. Based on the content of the article here are specific methodological improvements and further controls the authors should consider:

  1. Including a PRISMA flow diagram would greatly enhance transparency and reproducibility.
  2. Many studies do not stratify by baseline oxidative stress levels, yet these could affect outcomes significantly. Including this in future studies would improve the precision of intervention effects.
  3. The review does not address controls for diet, smoking, alcohol use, or physical activity, which are major modulators of oxidative stress and antioxidant status.

In summary, while the study is methodologically solid with its prospective, controlled, and randomized design, integrating these improvements would significantly enhance the internal and external validity, reduce bias, and provide stronger evidence for clinical recommendations.

Author Response

We greatly appreciate the time and effort invested in reviewing our manuscript, as well as the thoughtful suggestions provided. Your feedback has been instrumental in helping us improve the clarity, depth, and scientific rigor of the paper.

We have carefully considered all of your recommendations and have made corresponding revisions throughout the manuscript. We remain open to any additional suggestions that may help us further improve and refine our manuscript.

Once again, thank you for your valuable contribution to enhancing the quality of our work.

Comment:

  • Including a PRISMA flow diagram would greatly enhance transparency and reproducibility.

Response: Included, if it needs any enhancements, please mention.

Comment:

  • Many studies do not stratify by baseline oxidative stress levels, yet these could affect outcomes significantly. Including this in future studies would improve the precision of intervention effects.

Response: We agree that baseline oxidative stress levels could play a significant role in influencing intervention outcomes. We tried to include some other relevant studies. We also included a statement in the conclusions section about this topic.

Comment:

  • The review does not address controls for diet, smoking, alcohol use, or physical activity, which are major modulators of oxidative stress and antioxidant status

Response: Addressed now in section 3